# Wasserstein Generative Adversarial Network Based De-Blurring Using Perceptual Similarity

**Minsoo Hong and Yoonsik Choe \*** 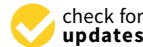

Department of Electrical & Electronic Engineering, Yonsei University, Seoul 03722, Korea; hms9110@naver.com
\* Correspondence: yschoe@yonsei.ac.kr; Tel.: +82-2-2123-2774

**Abstract:** The de-blurring of blurred images is one of the most important image processing methods and it can be used for the preprocessing step in many multimedia and computer vision applications. Recently, de-blurring methods have been performed by neural network methods, such as the generative adversarial network (GAN), which is a powerful generative network. Among many different types of GAN, the proposed method is performed using the Wasserstein generative adversarial network with gradient penalty (WGANGP). Since edge information is the most important factor in an image, the style loss function is applied to represent the perceptual information of the edge in order to preserve small edge information and capture its perceptual similarity. As a result, the proposed method improves the similarity between sharp and blurred images by minimizing the Wasserstein distance, and it captures well the perceptual similarity using the style loss function, considering the correlation of features in the convolutional neural network (CNN). To confirm the performance of the proposed method, three experiments are conducted using two datasets: the GOPRO Large and Kohler dataset. The optimal solutions are found by changing the parameter values experimentally. Consequently, the experiments depict that the proposed method achieves 0.98 higher performance in structural similarity (SSIM) and outperforms other de-blurring methods in the case of both datasets.

**Keywords:** deblurring; generative adversarial network; perceptual similarity; style information; Wasserstein distance

## 1. Introduction

De-blurring is one of the steadily studied techniques in image processing fields and aims to improve the sharpness of an image by eliminating blur noise. In Figure 1, the de-blurring method is used to transform the blurred image to the sharp image. To remove blur noise, many de-blurring techniques were researched, e.g., the blind deconvolution algorithm, bilateral and Wiener filter among others [1,2].

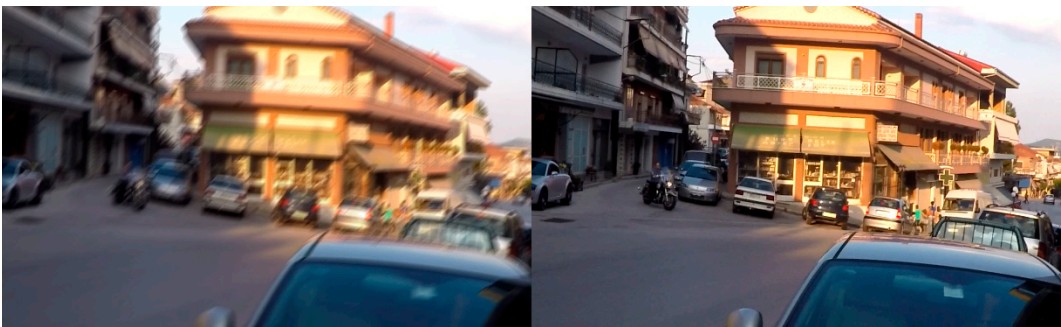

**Figure 1.** Blurred and sharp image of GOPRO dataset.

In recent years, deep learning methods have been adapted to image processing fields, showing better performance than the previous methods. Likewise, many researchers have applied deep learning methods, e.g., convolutional neural network (CNN) and generative adversarial network (GAN), in order to resolve blur noise [3–5]. Especially, since GAN shows high quality in generating texture information of images, this paper adapts the GAN-based approach to eliminate blur noise.

Typical GAN-based de-blurring methods [6] are processed by minimizing the difference of pixel values, which improves peak signal to noise ratio (PSNR). However, it cannot reflect the similarity for target images. Therefore, the proposed method uses the conditional GAN [7], which minimizes the difference of image distributions. In detail, proposed method adapts Wasserstein generative adversarial network with gradient penalty (WGANGP), which is based on conditional GAN. The Wasserstein generative adversarial network (WGAN) was proposed to stabilize the training of GAN by minimizing the Wasserstein distance of joint distributions instead of whole probability distributions [8]. Additionally, to improve the performance of WGAN, the gradient penalty term was proposed [9], known as a weight clipping method, in order to limit the gradient weight in range, and applying this term to the WGA can prevent the gradient vanishing and exploding problem.

By applying the content loss function, i.e., the procedure of extracting feature maps in CNN, to WGANGP, the improved de-blurring method was proposed to capture the perceptual information of images [10]. The content loss function is one of many perceptual loss functions and improves the similarity between the blurred and sharp images. Applying the content loss function to WGANGP can capture the perceptual information, such as color and spatial structure. However, it does not preserve the detailed edge information of image at the same time.

To preserve more edge information of the sharp image, another perceptual loss function called style loss function [11] is introduced. The style loss function extracts multiple feature maps in CNN and calculates by covariance matrix to capture more perceptual information and detailed edge. As a result, the style loss function captures higher similarity than the content loss function, shown with the similarity map in Figure 2. The figure shows that the boundary information between object and background is preserved in (b) better than (a).

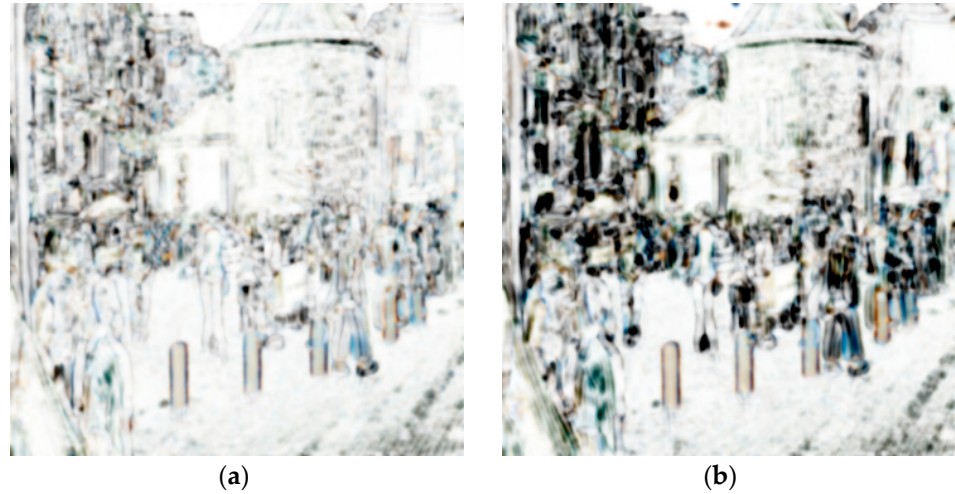

(**a**)  (**b**)

**Figure 2.** Similarity maps of (**a**) the content loss function and (**b**) the style loss function.

In this paper, the content loss function is replaced by the style loss function to preserve edge information and perceptual similarity. Substituting the style loss function to WGANGP can preserve the detailed edge since it can capture a large perceptual style and high level features. The objective function of network is proposed by combining the loss function of WGANGP and the style loss function. Therefore, the network generates de-blurred image while capturing the similarity of perceptual style and preserving detailed edge information.

In this work, it is experimentally shown that WGANGP with the style loss function can reconstruct edge-preserved de-blurring image and achieves high performance in structural similarity measure (SSIM). Also, a comparison experiment with different parameter values and the different locations of the feature map is performed to find the optimal solution. In Section 2, GAN, WGANGP and perceptual loss function are introduced and the problem definition to solve the blur noise is described. In Section 3, perceptual objective function and detail architecture of network are explained. In Section 4, reconstruct the de-blurred image using proposed method is reconstructed and the parameters and values to find optimal solution are analyzed, followed by the conclusion in Section 5.

## 2. Preliminaries

In this section, basic deep learning methods and the perceptual loss function for de-blurring are described. Section 2.1 introduces what GAN is and explains the objective function of GAN. It also describes the limitations of GAN and the ways to solve them. In 2.2, the Wasserstein GAN with a gradient penalty term is introduced for stabilizing the training process of GAN, followed by the descriptions of perceptual loss functions and how to calculate perceptual loss in 2.3. Finally 2.4 defines the issues in solving blur noise with the combination of WGAN-GP and perceptual loss function.

### 2.1. Generative Adversarial Network

The generative adversarial network (GAN) was proposed in 2014 by Goodfellow [6]. GAN is used for image generation task and is learned by a competition between two neural network models. Two neural network models are called the generator, G, and the discriminator, D. The goal of G is to generate images that D cannot distinguish from real images, and the goal of D is to differentiate real images and generated images. Therefore, the goal of GAN can be expressed as:

$$\min_{G} \max_{D} E_{x \sim P_r}[\log(D(x))] + E_{z \sim P_z}[\log(1 - D(G(z)))], \tag{1}$$

where $x$ is real data, $z$ is random Gaussian noise. $P_r$ and $P_z$ are the distributions of real images and generated images, respectively, and the base architecture of GAN is shown as Figure 3.

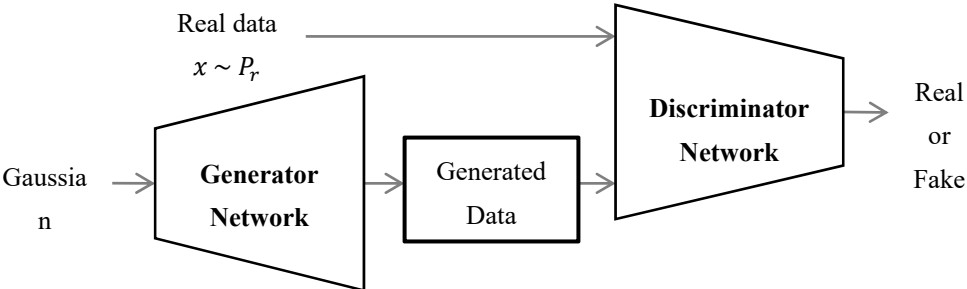

**Figure 3.** Architecture of GAN.

The objective function of GAN is a type of min-max problem, and it is hard to achieve Nash equilibrium. For this reason, GAN has an insecure training process and some problems, such as the gradient vanishing or exploding problem and mode collapse.

Figure 4 shows the optimal solution for GAN. The distribution of D should be a flat form, and the distribution of G should be the same as the real data, resulting in generating the best data. When D is perfect, GAN is guaranteed with $D(x) = 1$ and $D(G(z)) = 0$. Then, the objective function falls to zero and the loss updating gradient becomes zero, presenting the gradient vanishing problem. When D does not operates properly, inaccurate feedback is fed into G, and objective function cannot represent the reality. Also, if G does not learn the distribution of the entire training data and only a fraction of the training data is learned, the mode collapse problem occurs. In this case, G generates a limited diversity of samples or even the same sample, regardless of the input.

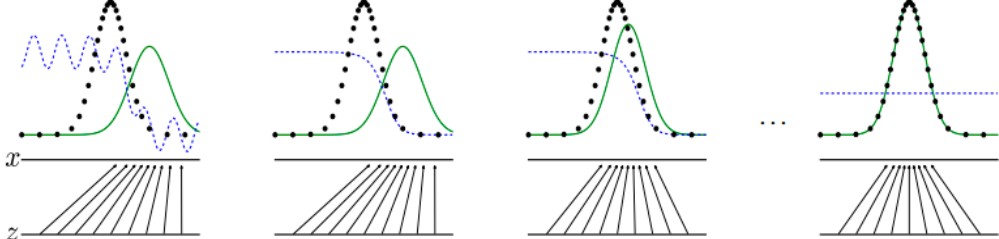

**Figure 4.** Distribution of discriminator (blue), generator (green) and real data (black) according to the learning process.

To avoid the above problems and make stable training process, Wasserstein GAN was proposed [8]. It has a new objective function derived from the Wasserstein distance, which is a measure of the distance between two joint probability distributions. A detailed description of WGAN will be mentioned in 2.2.

*2.2. Wasserstein GAN with Gradient Penalty*

The Wasserstein GAN (WGAN) was proposed by Arjovsky in 2017 [8], which uses Wasserstein distance to measure the distance between two joint probability distributions. Wasserstein distance is expressed as Equation (2):

$$W\big(P_r, P_g\big) = \inf_{\gamma \in \Pi(P_r, P_g)} E_{(x,y)\sim\gamma}\Big[\big\|x - y\big\|\Big], \tag{2}$$

where $\Pi\big(P_r, P_g\big)$ denotes the set of all joint distributions $\gamma(x, y)$, and $\gamma(x, y)$ represents the distance for transforming the distribution $P_r$ into the distribution $P_g$.

The Wasserstein distance is a weaker metric than the others, such as total variance (TV), Kullback-Leibler divergence (KL) and Jensen-Shannon divergence (JS). Minimizing the objective function of GAN is equal to minimizing JS divergence and JS divergence, determined that two probability distributions $P_r$, $P_g$ are completely different when measured in different areas. In other words, they look harshly different in two probability distributions. In GAN, this reason can cause the discriminator to fail to learn. Therefore, the Wasserstein distance, which is flexible and focuses on convergence, is applied to train the process of GAN.

The reason why Wasserstein distance is a weak metric is shown in Figure 5.

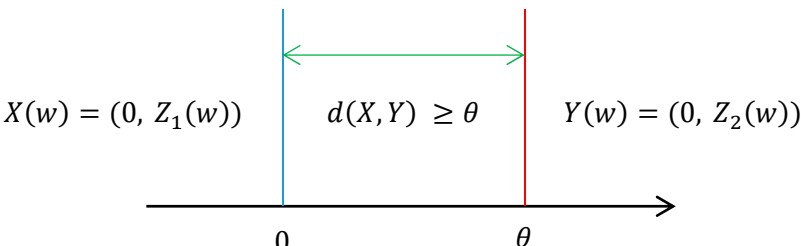

**Figure 5.** Example of two probability distributions.

Where $X$ and $Y$ are random variables mapped as $X \sim P_0$, $Y \sim P_\theta$, respectively, and $d(X, Y)$ is distance between $X$ and $Y$. Here, $d(X, Y)$ is calculated as follows:

$$d(X, Y) = \big(|\theta - 0|^2 + \big|Z_1(w) - Z_2(w)\big|\big)^{\frac{1}{2}} \geq |\theta|. \tag{3}$$

The expected value of $d(X, Y)$ is equal to or greater than $\theta$ with any joint probability distribution $\gamma$:

$$E^\gamma[d(X, Y)] \geq E^\gamma[|\theta|] = |\theta|. \tag{4}$$

When $Z_1$ is equal to $Z_2$, the expected value of $d(X, Y)$ becomes $|\theta|$. Then, the desired conclusion is achieved as the following Equation (5) and Figure 6.

$$W(P_0, P_\theta) = |\theta| \tag{5}$$

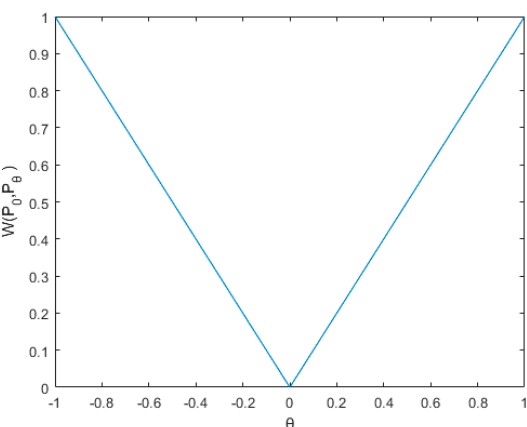

**Figure 6.** Graph of Equation (5).

As a result, the objective function of WGAN can be expressed by the Kanotorovich-Rubinstein duality [12]:

$$\min_{G} \max_{D \in L} \mathop{E}_{x \sim P_r} [D(x)] - \mathop{E}_{x \sim P_g} [D(G(\widetilde{x}))], \tag{6}$$

$$\lambda \mathop{E}_{\widetilde{x} \sim P_{\widetilde{x}}} [(\left\| \nabla_{\widetilde{x}} D(\widetilde{x}) \right\|_2 - 1)^2]. \tag{7}$$

Gradient penalty term limits gradient weight to range $[-c, c]$, where c is the threshold. Applying a gradient penalty term to WGAN can prevent gradient vanishing and the exploding problem.

### 2.3. Perceptual Loss Function

The perceptual loss function is used in image transfer tasks by extracting representations of a feature map. There are two types of perceptual losses which are the content loss function and the style loss function. Basic merit of the perceptual loss function is the ability to extract feature responses in layers of CNN. As the network goes deeper, the input image is changed to be representations of features, not of pixel values. That is because features of the high layer have large receptive fields, and it represents the actual content of the image and spatial structure. Therefore, the high layer of the network can capture high-level contents and perceptual information, as in Figure 7, and the feature response of high layers is called the content representation.

To minimize the difference of content representation between the input and target images, the content loss function was proposed in [10], and it is represented as follows:

$$\mathcal{L}^{\varnothing,j}_{\text{content}} (x, \hat{x}) = \frac{1}{C_j H_j W_j} \left\| \varnothing_j(\hat{x}) - \varnothing_j(x) \right\|_2^2, \tag{8}$$

where $x$, $\hat{x}$, $\varnothing_j$ and $C_j \times H_j \times W_j$ are input image, target image, feature map of j-th layer and size of feature map, respectively. Usually, feature map is extracted in pre-trained networks by ImageNet datasets, such as VGGNet16 or VGGNet19. The content loss function is calculated by mean square error.

In the de-blurring task, using the content loss function for a network encourages the output image to be perceptually similar to the sharp image, since it can capture the content of sharpness information. To capture more perceptual similarity and texture information, the style loss function was proposed in [11]. The style loss function is similar to the content loss function, which uses feature responses in layer of network. However, the style loss function consists of the correlations between different

feature responses, and it extracts multiple feature maps of network. By considering the correlation of features, the style loss function obtains multi-scale representations. Correlation of features is given by a Gram matrix, which is a covariance matrix that represents the distribution of an image, and the Gram matrix is the inner product between different vectored feature maps. The Gram matrix for style loss is expressed as follows:

$$G_j^{\varnothing}(x)_{c,c'} = \frac{1}{C_j H_j W_j} \sum_{h=1}^{H} \sum_{w=1}^{W} \varnothing_j(x)_{h,w,c} \varnothing_j(x)_{h,w,c'} \tag{9}$$

where $G_j^{\varnothing}(x)$, $\varnothing_j(x)_{h,w,c}$ and $\varnothing_j(x)_{h,w,c'}$, and $C_j \times H_j \times W_j$ are the Gram matrix of feature map in j-th layer, different feature maps with different channels, and the size of feature map, respectively.

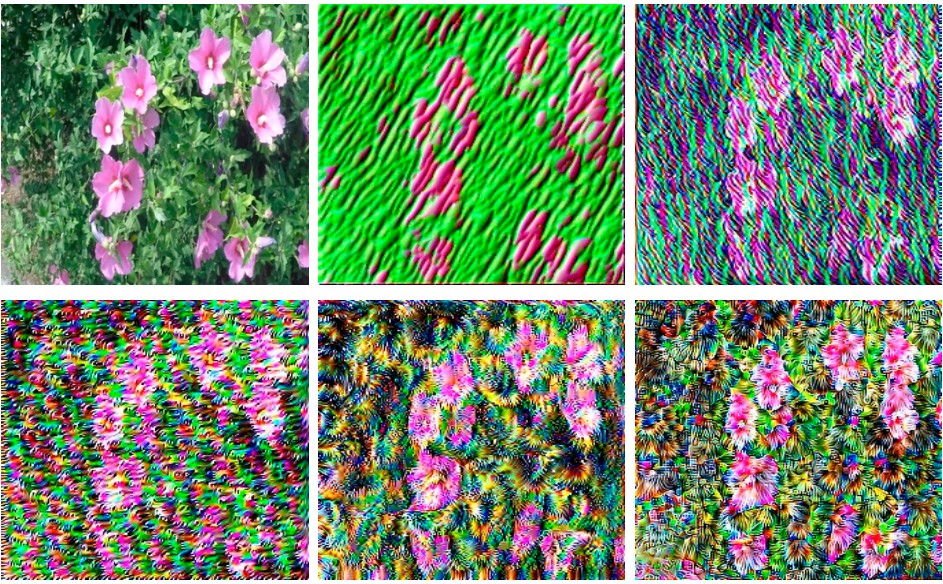

**Figure 7.** Feature response of layers in CNN. First row is original image, conv_2 layer and conv_4 layer. Second row is conv_6 layer, conv_9 layer and conv_12 layer.

For the efficient calculation of the Gram matrix, feature maps are reshaped by changing the size of $C_j \times H_j \times W_j$ to $C_j \times H_j W_j$. After reshaping, the style loss function is calculated using difference between the Gram matrices of the target and the output images. It is minimized by using the Frobenius norm, and the formula of the style loss function is as follows:

$$\mathcal{L}_{\text{style}}^{\varnothing,j} = \frac{1}{C_j H_j W_j} \left\| G_j^{\varnothing}(\hat{x}) - G_j^{\varnothing}(x) \right\|_F^2. \tag{10}$$

As a result, applying the syle loss function to WGAN can preserve more detailed edge information and capture the similarity of perceptual styles.

### 2.4. Problem Definition

Figures 8 and 9 show experimental results of WGAN-GP with the content loss function. These experiments are performed with the GOPRO dataset. The goal of this network is to reconstruct the de-blurred image, that is similar to the sharp image, with the use of the blur image. Blurred and sharp images are fed to the network, with the network learning the perceptual content and similarity of the sharp image.

In Figure 8, trained network with WGAN-GP and content loss function generates de-blurred image, which has perceptual content information. It results in good performance in PSNR and outputs de-blurred image by eliminating blur noise. However, a problem exists when the network reconstructs

image. As shown in extending the sub-part of generated images in Figure 9, block effects occur in objects and backgrounds. This is because the content loss function uses only one feature map in CNN. That is, using a single feature map to capture perceptual information is not enough to represent micro-edges in small objects, such as a leaf, branch, etc.

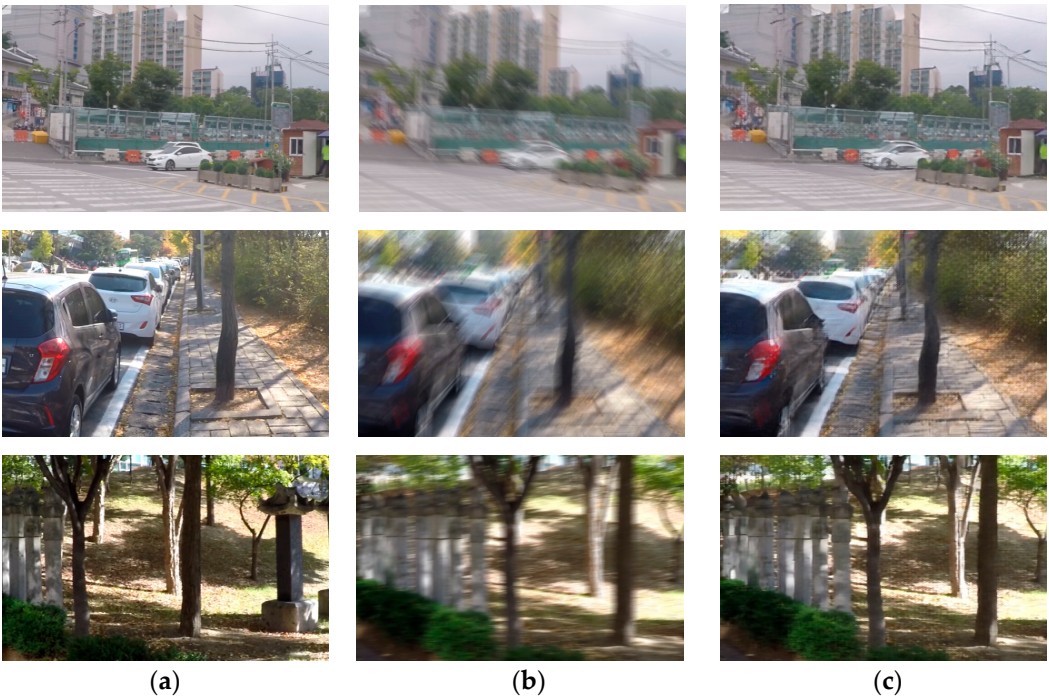

|        (**a**)        |        (**b**)        |        (**c**)        |

**Figure 8.** Example images of WGAN-GP with content loss function: (**a**) Sharp image. (**b**) Blurred image (**c**) Reconstruct image.

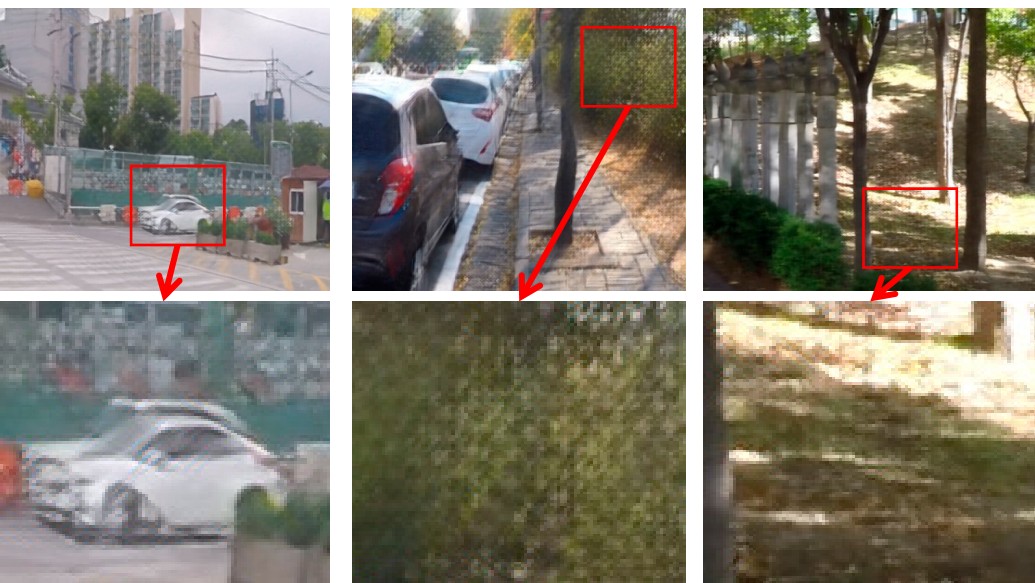

**Figure 9.** Extending part of example images.

In this regard, to improve the sharpness of the output image, the content loss function is replaced by the style loss function in order to extract multiple feature maps in CNN. Therefore, adapting the style loss function increases feature complexity and the size of the receptive field. Moreover, it considers the correlation between different feature maps of layers.

The next section explains the addition of the style loss function to WGAN-GP and analyzes how it is calculated. Finally, the de-blurring network that can generate the realistic de-blurred image and show a high similarity between output and sharp image is proposed.

## 3. Proposed Algorithm

This section describes the proposed method and detailed composition to reconstruct de-blurred image. First, perceptual style by extracting multiple feature map in high layer is described. Then, the total loss function which combines WGAN-GP and the style loss function follows. Next, the way to minimize the total loss function is introduced. Finally, the architecture of the de-blurring network is depicted.

### 3.1. Multi-Scale Representation of Perceptual Style

To capture the perceptual style of the sharp image, style loss function is used in the proposed method. As mentioned in Section 2, applying the style loss function can increase the similarity between the generated and sharp images by minimizing the difference of distributions. It is similar to the content loss function, but the style loss function extracts multiple feature maps in CNN to increase feature complexity and receptive field.

Usually, feature maps of layers in VGG16 are used to capture similarity and perceptual style. VGG16 [13] is a type of CNN, which has deep convolutional layers and is trained on ImageNet. The architecture of VGG 16 is shown in Figure 10.

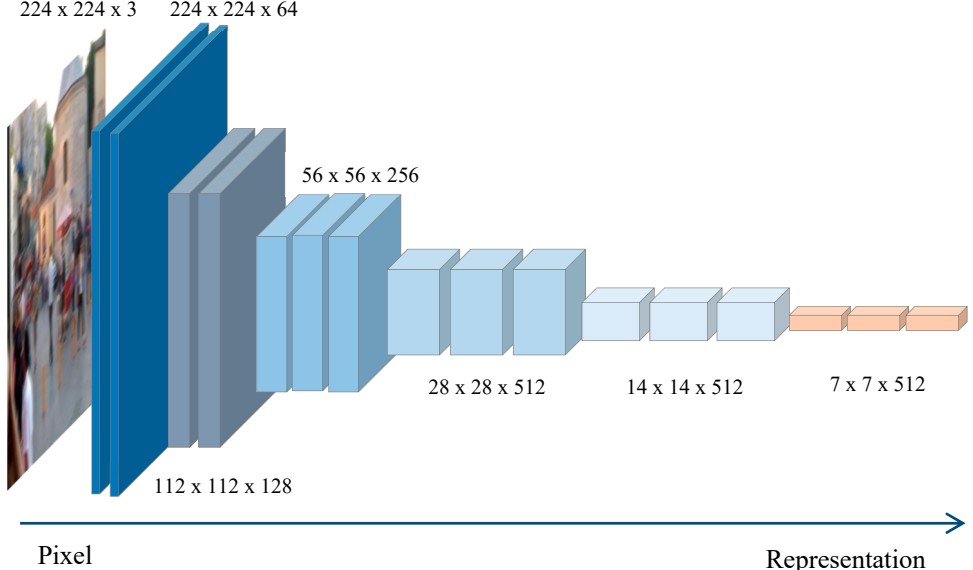

**Figure 10.** Architecture of VGG16 network.

VGG16 adapts a 3 × 3 filter convolution repeatedly to capture the representation of the image. Then feature map of the high layer includes high-level features that obtain a high complexity representation with large receptive field.

When the network goes deeper, the input image is transformed to representation values, and the size of feature maps is reduced as the depth is increased. To take multi-scale representation, five feature maps, which are in different shapes, are extracted. Also, feature maps of the high layer, which contain high non-linearity, show better performance than feature maps of low layer. Feature maps of the low layer are not sufficient to express the perceptual style of an image because they contain low-level features.

The reconstructed image using low-level features is shown in Figure 11. Low-level feature cannot capture texture, color and style because the feature is too simple and takes small receptive field.

The reconstructed image has black space owing that it does not fully express the style and texture of the image. As a result, to reconstruct the de-blurring image and preserve the content with perceptual style, this paper proposes extracting five feature maps in high layer, which have enough perceptual style and edge information.

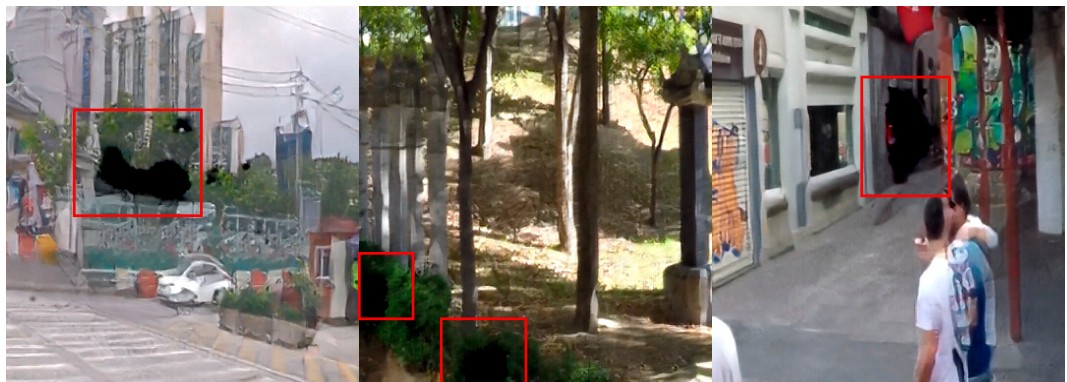

**Figure 11.** Generated image using low-level feature.

### 3.2. Total Loss Function of Network

The objective of the proposed method is to generate de-blurring image from the blurred image, $I_B$, by extracting the perceptual style information from the sharp image, $I_S$. The network learns the similarity of the sharp image, and it reconstructs the de-blurred image from the blurred image by using micro-edge information and perceptual similarity. The similarity can be improved by minimizing the difference of distribution between $I_S$ and $I_B$. As a result, well-trained generator, component of network, generates output image that include edge information.

To train the network, a discriminator and generator are alternately learned by blurred and sharp images. The learning direction of the discriminator in the WGAN-GP is decided to find the optimal solution by minimizing the Wasserstein distance between joint distributions of the generated and sharp images. The loss function of the discriminator is as follows:

$$\mathcal{L}_\mathrm{D} = \sum_{n=1}^{N} [L(I_s) - L(G_\theta(I_B)) \; + \; \lambda \big( \big\| \nabla L(\epsilon I_S + (1 - \epsilon) G_\theta(I_B)) \big\|_2 - 1 \big)^2 ], \tag{11}$$

where $L$ is a differentiable function, which is 1-Lipschitiz only if it has gradients with norm at most 1 everywhere. $G$ and $\epsilon$ are generator and random number U[0,1], respectively, and $\lambda$ is the gradient penalty coefficient set to 10. The gradient penalty term penalizes the network if the gradient norm moves away from its target norm value 1.

The discriminator is trained first, along with the training of a generator, to generate the de-blurring image. To make generator reconstruct images with perceptual style, the total objective function is obtained by the combination of generator loss function and style loss as shown in Equation (12):

$$\mathcal{L} = \mathcal{L}_\mathrm{G} + \lambda \cdot \mathcal{L}_{style}, \tag{12}$$

where $\mathcal{L}_\mathrm{G}$ is generator loss function of WGAN-GP, and $\mathcal{L}_{style}$ is style loss function to capture the perceptual style from sharp image. Also, $\lambda$ is a constant value of weight parameter that determines how much perceptual style should be adapted. In more detail, $\mathcal{L}_\mathrm{G}$ is calculated by WGAN-GP method.

$$\mathcal{L}_\mathrm{G} = \sum_{n=1}^{N} -L(G_\theta(I_B)), \tag{13}$$

where $L$ is the critic function as mentioned above and $G$ is the generator. Moreover, $\theta$ is parameter of the network which minimizes loss function:

$$\mathcal{L}_{style} = \sum_{m=1}^{M} \frac{1}{C_j H_j W_j} \left\| G_j^\varnothing \left( G_\theta(I_B) \right) - G_j^\varnothing (I_S) \right\|_{F'}^2 \tag{14}$$

where $M$ is the number of feature map which is extracted from VGG16. Proposed method uses five feature maps of high layer in VGG16. Using fewer feature maps cannot express perceptual style, and if more feature maps are used, more calculation and lot of training time are needed. In summary, a novel combined loss function consisting of WGAN-GP and style loss function is proposed to reconstruct edge preserved de-blurring image with perceptual style.

### 3.3. Network Architecture

The architecture of the generator is shown in Figure 12, where the generator is a CNN based on residual network. It is composed of three convolutional layers, nine residual block (Resblock) [14] and two transposed convolutional layers. First, to encode the characteristics of images, convolutional layer, instance normalization layer [15] and ReLU activation layer [16] are designed in front of the network. The size of an image is decreased and the depth of feature map is increased. After that, nine residual blocks are connected behind the convolutional layer to increase feature complexity. Each Resblock consists of a convolutional layer with dropout regularization [17], instance normalization layer and ReLU activation layer.

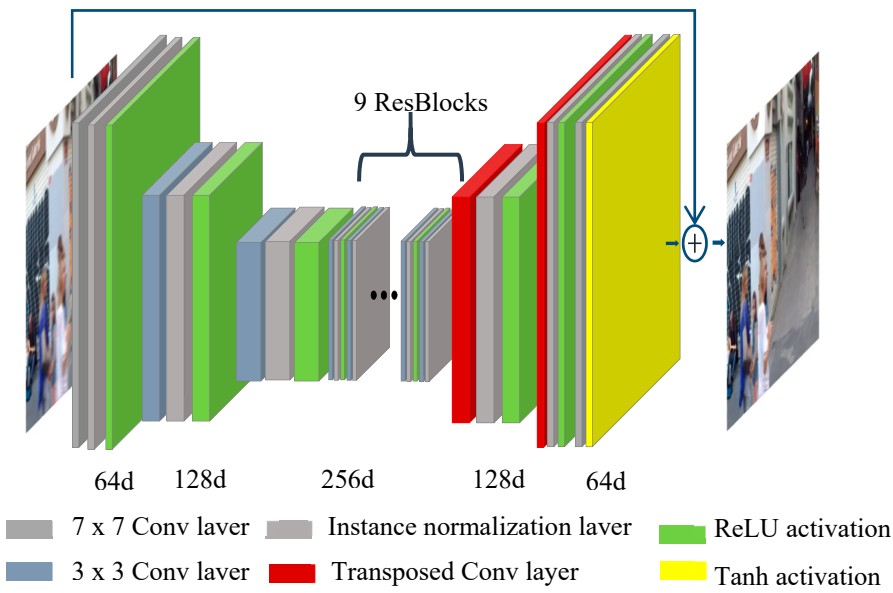

**Figure 12.** Architecture of generator in WGAN-GP.

In the back of the network, the transposed convolutional layer is attached to reshape feature maps to generate output images by up-sampling, and Tanh activation is applied in last convolutional layer.

The architecture of the discriminator is shown in Figure 13, having the same architecture as Patch-GAN [18,19]. Patch GAN was proposed to classify whether each N × N patch in an image is real or fake.

The discriminator consists of five convolutional layers and three instance normalization layers. Unlike the generator, LeakyReLU [20] activation layer is applied to the convolutional layers.

Figure 14 shows architecture of the entire network. The network is based on conditional GAN, and blurred and sharp images are the input for the network. The generator produces the estimate of

de-blurring image, and the discriminator differentiates the sharp and de-blurring image. Moreover, style loss function is added to generator loss function to capture perceptual style.

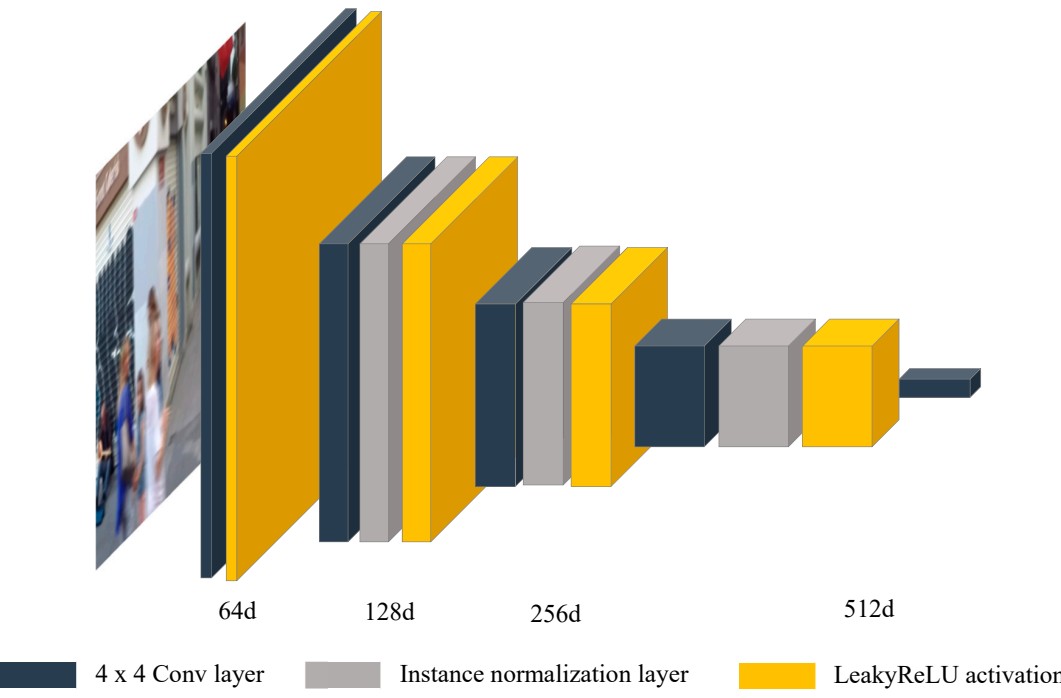

**Figure 13.** Architecture of discriminator in WGAN-GP.

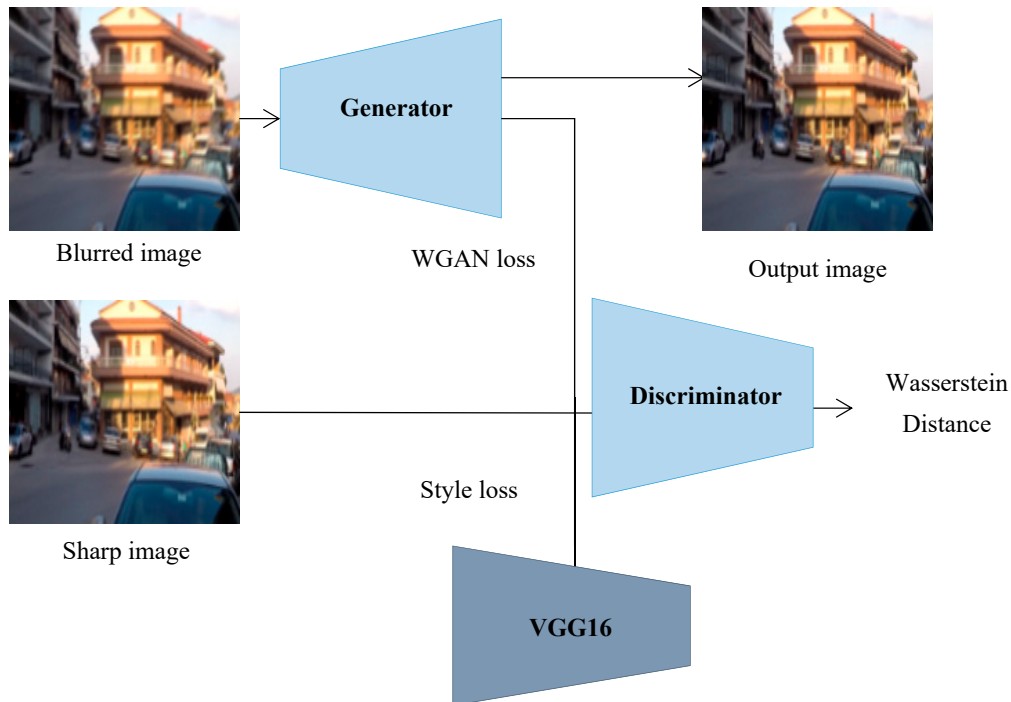

**Figure 14.** Configuration diagram for the entire network.

## 4. Experimental Results

Section 4 describes the experimental environment and experiment results for two datasets. Proposed method shows better performances in de-blurring, compared to the filter based methods

and content based method. Analysis of experiments is also carried out according to the parameters, followed by the comparisons between the proposed method and conventional methods.

### 4.1. Experimental Environments

All of experiments are implemented in Linux using Python and Pytorch as deep learning library. Training is performed on Intel Core i7-6700K CPU @4.00GHz, 32 GB of RAM, Geforce GTX-1080Ti, and it takes 7 days to finish training the network. The network is trained on GOPRO Large dataset [7], which consists of 2013 training data and 1111 test data. Input for the network is a pair of blurred and sharp images, and image resolution is resized from 1280 × 720 to 640 × 360 for training.

In addition, 48 test data of Kohler dataset are used for testing the network. Kohler dataset [21] is a blur-sharp dataset that applies twelve different blur kernels to each of the four images. Image resolution is resized from 800 × 800 to 400 × 400. Blur kernels are shown in Figure 15.

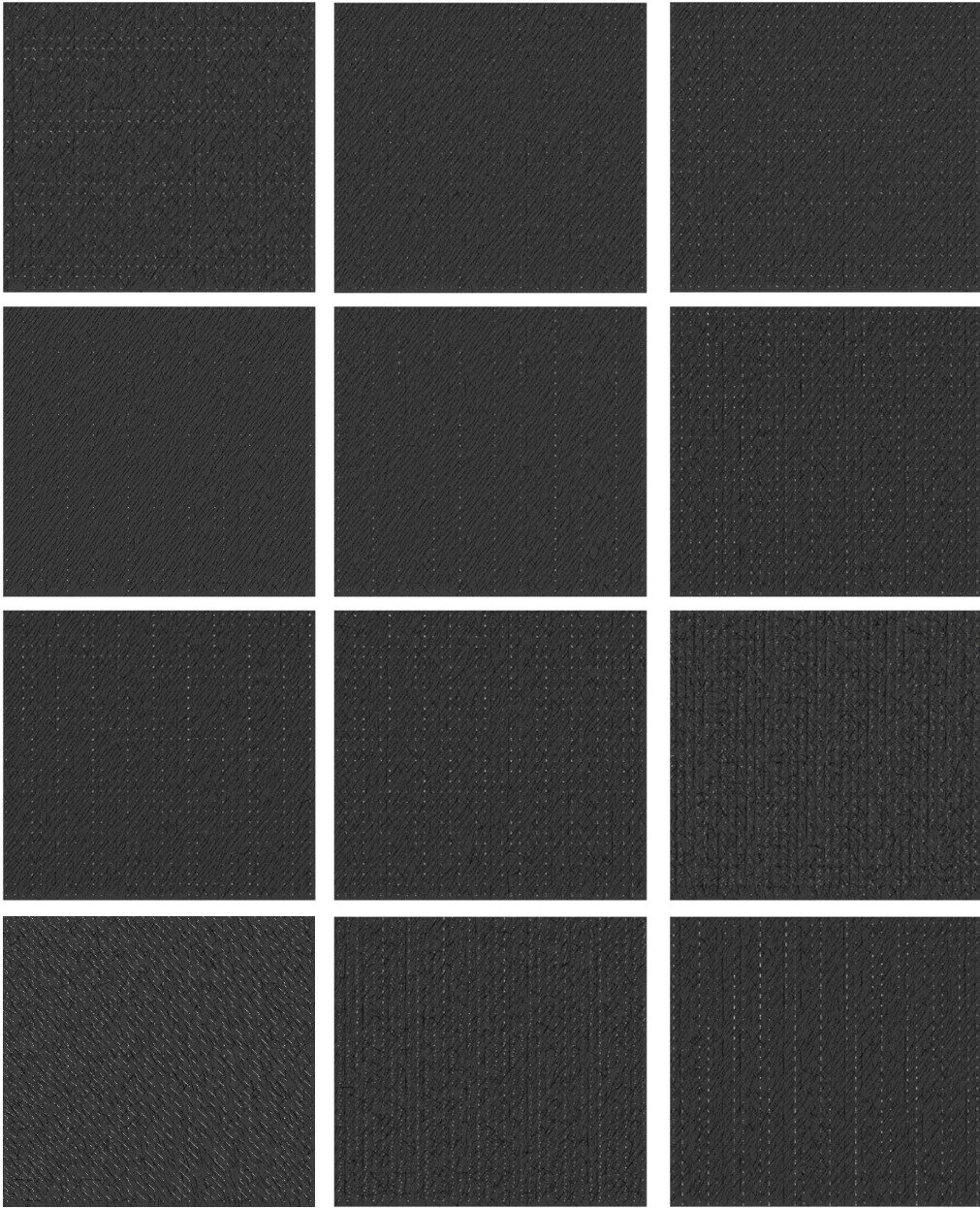

**Figure 15.** Twelve different blur kernels in Kohler dataset.

### 4.2. Compare with Filter based Methods

In this section, the results are compared from proposed method and other filter-based methods, such as wiener and bilateral filter, on two datasets. Wiener and bilateral filters are typical methods for de-blurring. Filter-based methods are applied to blurred image to eliminate the blur noise in the image. However, they do not reflect non-linearity of data and perceptual style information because they cannot learn the features of the image.

The results of the comparison of the proposed method with other filter-based methods using two different datasets are given in Tables 1 and 2. The performances of methods are evaluated by means of peak signal to noise ratio (PNSR) and structural similarity measure (SSIM). The results of PSNR and SSIM on GOPRO Large dataset are measured in the top 50 averages, and the result on Kohler dataset are measured in the top 5 averages.

**Table 1.** PSNR and SSIM on GOPRO Large dataset.

| Method | Compare Proposed Method to Filter Method | |
|---|---|---|
| | PSNR | SSIM |
| Bilateral filter [1] | 26.67 | 0.93 |
| Wiener filter [2] | 28.58 | 0.92 |
| Proposed method | **33.29** | **0.98** |

**Table 2.** PSNR and SSIM on Kohler dataset.

| Method | Compare Proposed Method to Filter Method | |
|---|---|---|
| | PSNR | SSIM |
| Bilateral filter [1] | **25.24** | 0.84 |
| Wiener filter [2] | 24.51 | 0.84 |
| Proposed method | 23.29 | **0.86** |

As a result, the proposed method, which minimizes distributions between two images, shows better results in PSNR and SSIM than other filter-based methods, which minimizes pixel-based differences between two images.

Figures 16 and 17 show the output images of filter-based methods and proposed method using GOPRO Large dataset. With the proposed method, the blur noise in the image is removed, and the output images are similar to the sharp image. Also, the output images using the Kohler dataset are shown in Figure 18. This dataset is made by applying artificial blur noise rather than natural blur noise. The proposed network reconstructs de-blurred image better than the filter-based methods in SSIM. This is because the network captures the perceptual style and complex features to reconstruct the de-blurred image, and the filter-based methods and proposed method have a different form to minimize value.

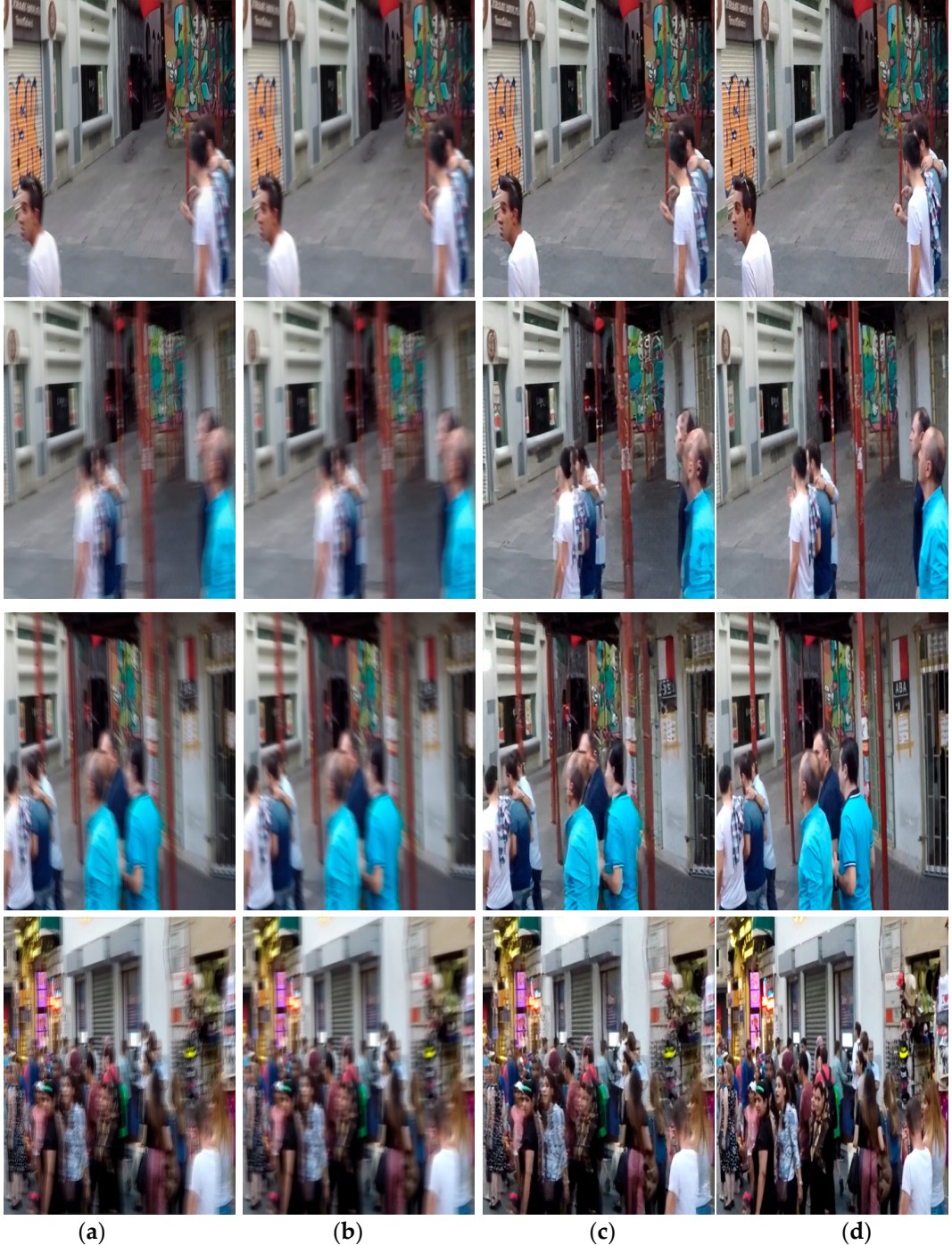

**Figure 16.** Object images from comparing proposed method with filter-based methods: (**a**) Wiener filter (**b**) Bilateral filter (**c**) Proposed method (**d**) Original sharp image.

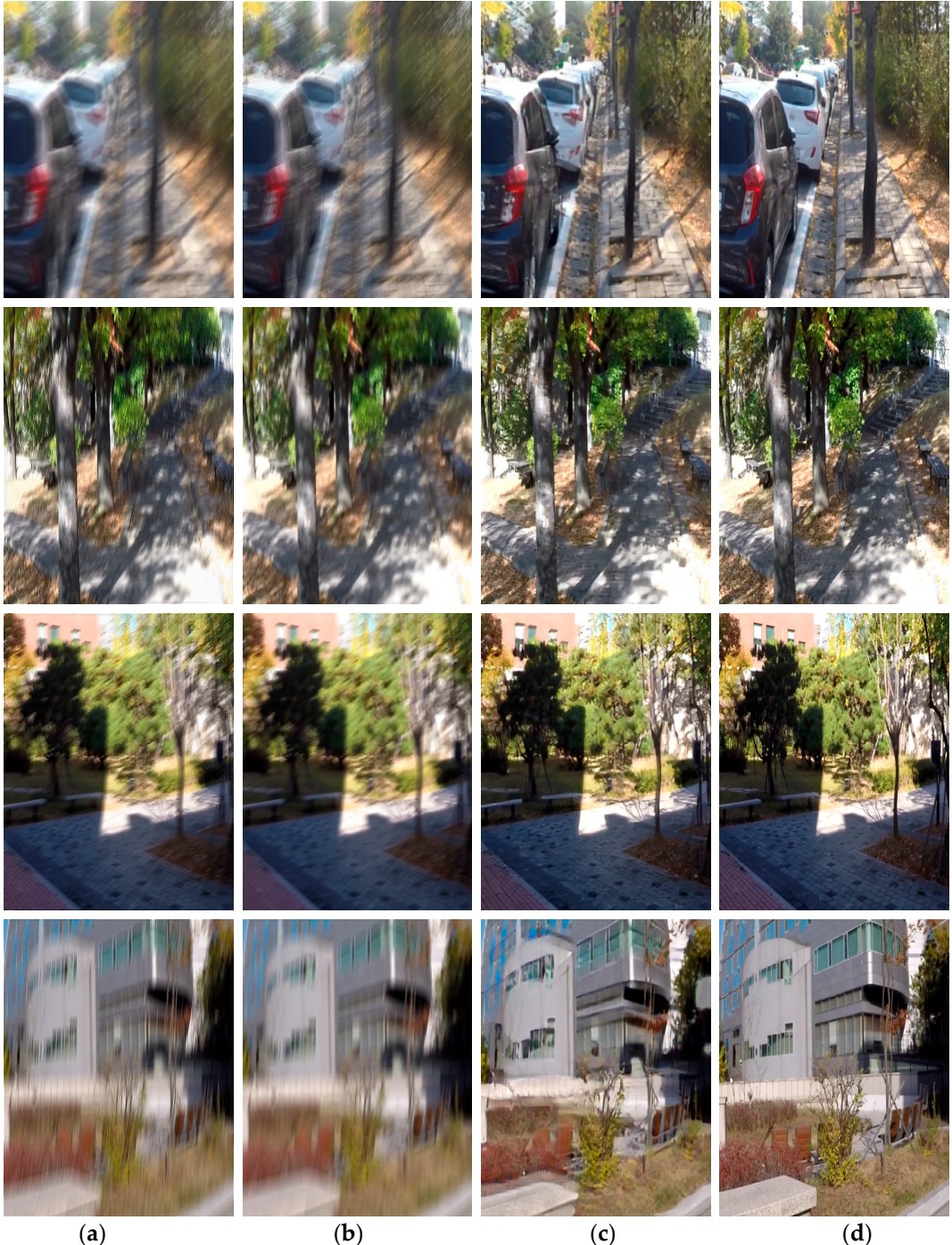

**Figure 17.** Background images from comparing proposed method with filter-based methods: (**a**) Wiener filter (**b**) Bilateral filter (**c**) Proposed method (**d**) Original sharp image.

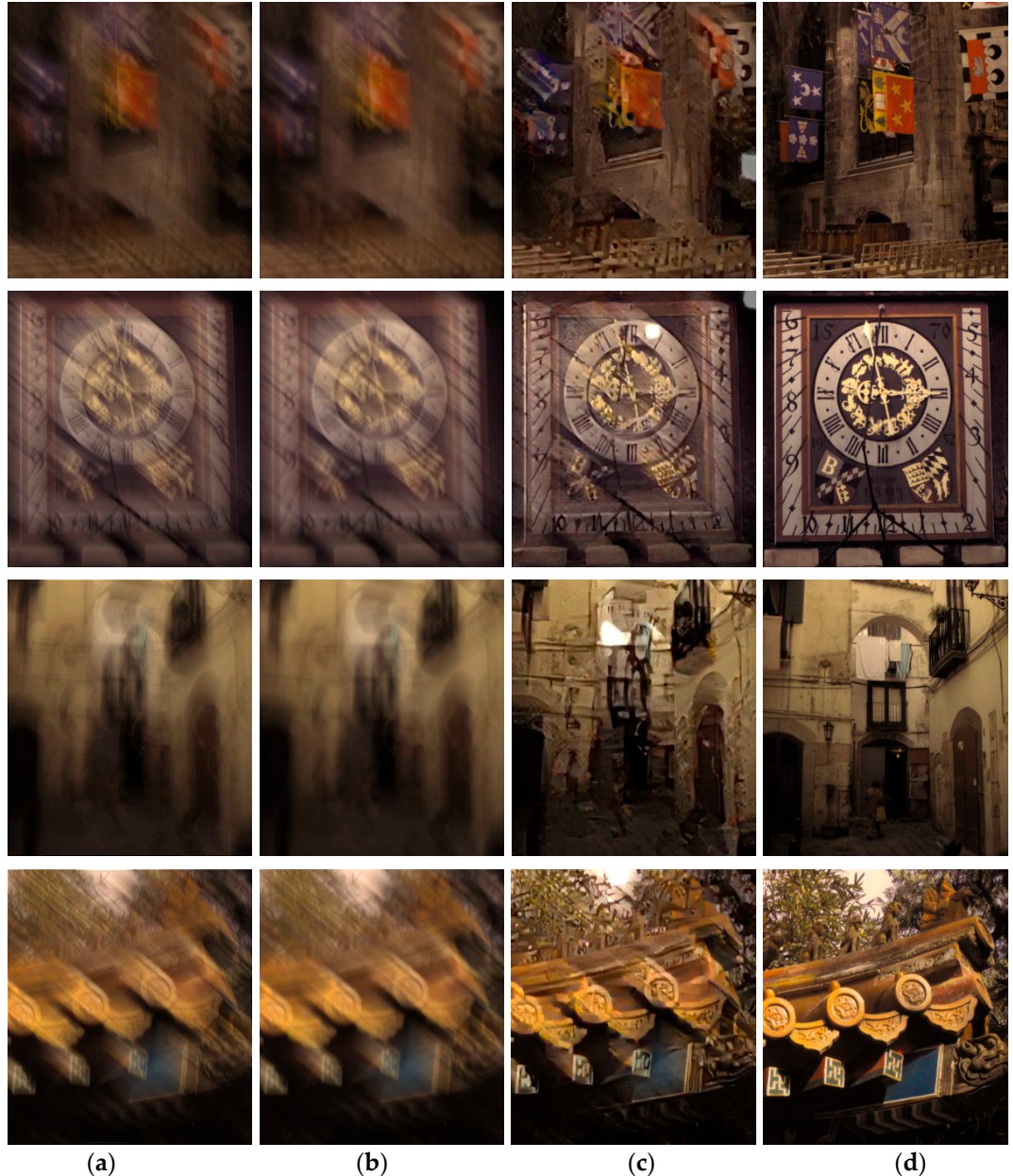

|  (**a**)  |  (**b**)  |  (**c**)  |  (**d**)  |

**Figure 18.** Output images from comparing proposed method with filter-based methods on Kohler dataset: (**a**) Wiener filter (**b**) Bilateral filter (**c**) Proposed method (**d**) Original sharp image.

### 4.3. Compare with Content Loss Function

In this section, the proposed method is compared with WGAN-GP using content loss. The proposed method using style loss, which takes multi-scale feature maps and considers the correlation of features, shows better results in PSNR and SSIM than method using content loss with only one feature map. Also, the results of PSNR and SSIM are shown in Tables 3 and 4.

**Table 3.** PSNR and SSIM on GOPRO Large dataset.

| Method | Compare Proposed Method to WGAN-GP with Content Loss | |
| --- | --- | --- |
|  | PSNR | SSIM |
| WGAN-GP with content loss [5] | 32.96 | 0.97 |
| Proposed method | **33.29** | **0.98** |

**Table 4.** PSNR and SSIM on Kohler dataset.

| Method | Compare Proposed Method to WGAN-GP with Content Loss | |
| --- | --- | --- |
| | PSNR | SSIM |
| WGAN-GP with content loss [5] | 23.24 | 0.80 |
| Proposed method | **23.29** | **0.86** |

Figures 19 and 20 are generated images from proposed method and WGAN-GP with content loss on GOPRO Large dataset, and Figure 21 is generated image on Kohler dataset. Both methods generate sharp images since they use feature map in VGG16; however, output image of WGAN-GP with content loss shows block effect which looks like a check pattern. With the de-blurred image of proposed method using style loss, the block effect is removed, and detailed edge in sub-part of image, e.g., branch, leaf and background, are preserved. Consequently, proposed method generates de-blurred images that are more natural to human eyes.

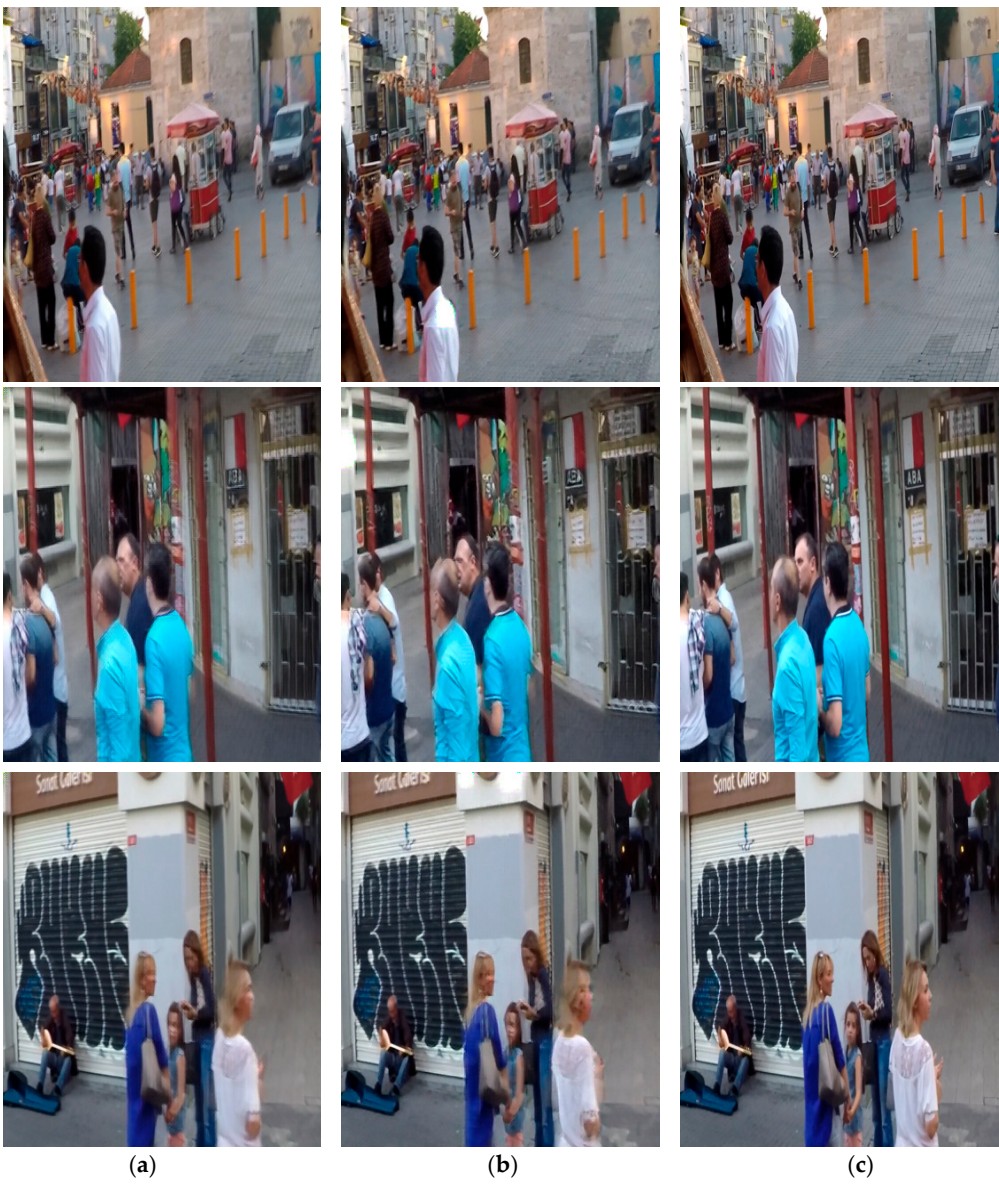

(**a**)            (**b**)            (**c**)

**Figure 19.** Object images from comparing proposed method with WGAN-GP using content-based methods: (**a**) WGAN with content loss (**b**) Proposed method (**c**) Original sharp image.

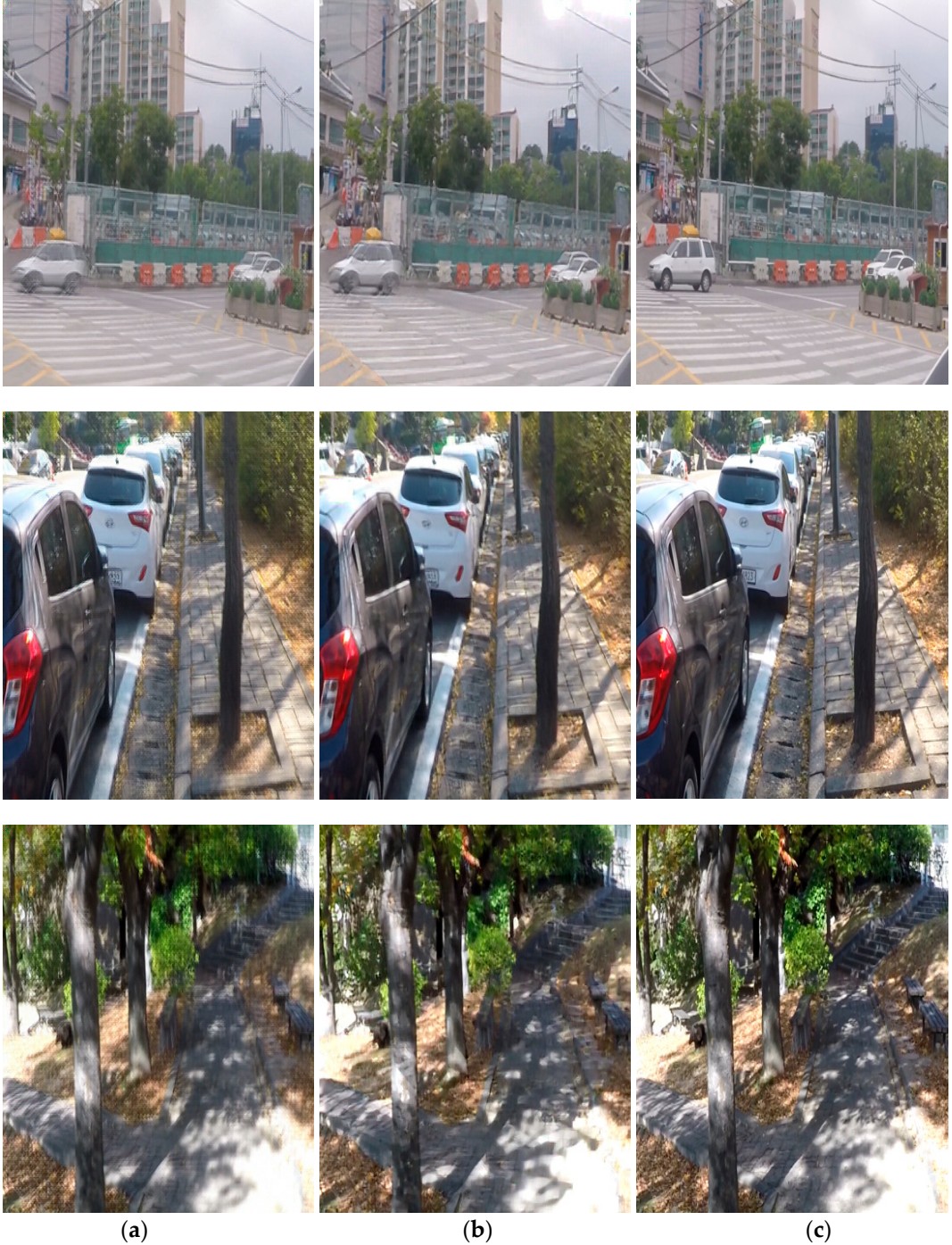

(**a**) (**b**) (**c**)

**Figure 20.** Background images from comparing proposed method with WGAN-GP using content -based methods: (**a**) WGAN with content loss (**b**) Proposed method (**c**) Original sharp image.

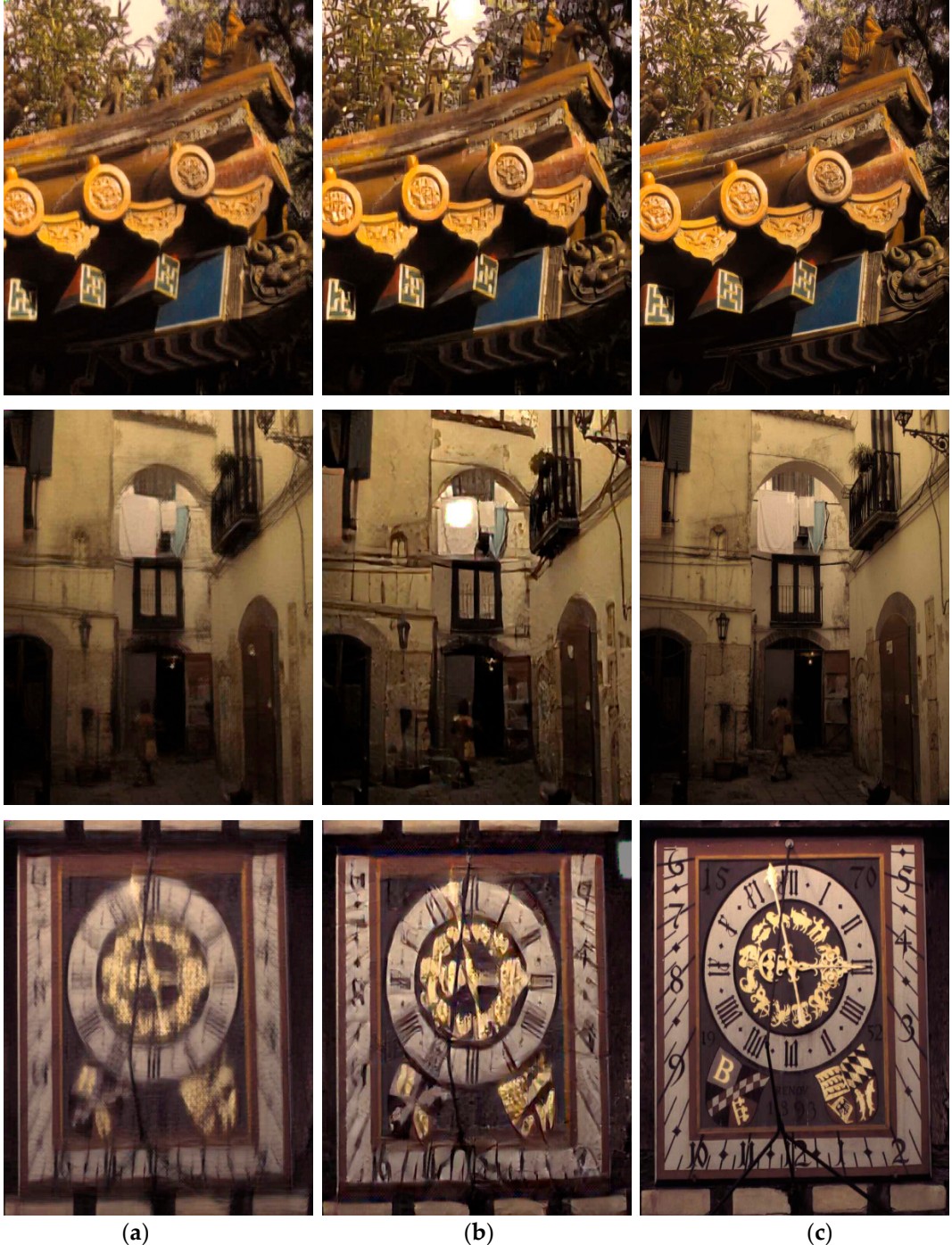

**Figure 21.** Generated images from comparing proposed method with WGAN-GP using content loss method on Kohler dataset: (**a**) WGAN with content loss (**b**) Proposed method (**c**) Original sharp image.

### 4.4. Experiments for Optimal Solution

Finding the optimal solution means that network generates a high performance of de-blurred images. Various experiments are conducted on different parameters. In the first experiment, five different feature maps of layers in VGG16 [10] are used for calculating style loss in three different ways.

1.  Using relu2_1, relu2_2, relu3_1, relu3_2 and relu3_3 in low layers.
2.  Using relu4_1, relu4_2, relu4_3, relu4_4 and relu5_1 layer in high layers.
3.  Combine low layer and high layers, which consist of relu2_2, relu3_1, relu4_3, relu4_4 and relu5_1.

Tables 5 and 6 show results of PSNR and SSIM from different locations of feature map in style loss. They show that feature map of high layer contains more detailed edge information. This is because the features of high layers are obtained from large receptive fields. Therefore, to reconstruct the de-blurred image, it is the best to use a high layer feature map in order to capture detailed edge and texture information at the same time.

**Table 5.** PSNR and SSIM of different layer on GOPRO Large dataset.

| Method | Comparing Between Different Layers | | |
| --- | --- | --- | --- |
| | Low Layer | Combine Layer | High Layer |
| PSNR | 29.69 | 30.02 | **33.29** |
| SSIM | 0.94 | 0.96 | **0.98** |

**Table 6.** PSNR and SSIM of different layer on Kohler dataset.

| Method | Comparing Between Different Layers | | |
| --- | --- | --- | --- |
| | Low Layer | Combine Layer | High Layer |
| PSNR | 22.45 | 21.59 | **23.29** |
| SSIM | 0.77 | 0.78 | **0.86** |

In Figures 22 and 23, reconstruction images from extracting low layers has a black hole in the image. As mentioned above, low-level features cannot express texture and edge information, meaning that they do not have enough information to restore the image. As a result, the reconstructed image from extracting high layers represents high quality texture and edge property.

The second experiment to find the optimal solution is performed by changing lambda values, which are the ratios of style loss and WGAN-GP loss. This is experimented in three cases, which are $\lambda = 10$, $\lambda = 100$, and $\lambda = 1000$. Output images are shown in Figures 24–26. When $\lambda$ is 10, generator cannot express enough style information for de-blurred image. This case still has blur noise, and it does not contain style information. Also, when $\lambda$ is 1000, too much style information is applied to the resulting image, causing the loss of edge components, and the boundaries of the generated images have been crushed. Experimentally, using the value of $\lambda$ as 100 can generate the best de-blurred image which contains style information and edge components, and de-blurred images look more natural to the human vision.

Results of PSNR and SSIM are shown in Tables 7 and 8. Table 7 shows the results using GOPRO Large dataset and Table 8 using Kohler dataset. As mentioned above, if $\lambda$ is too small or too large, its performance in PSNR and SSIM will be dropped, and the generated images do not reflect the edge information because the network does not utilize the appropriate style information. Using 100 for $\lambda$ gives the best possible performance and can reconstruct the de-blurred image with style and edge information.

**Table 7.** PSNR and SSIM of different lambda on GOPRO Large dataset.

| Method | Comparing Between Different Layers | | |
| --- | --- | --- | --- |
| | $\lambda=10$ | $\lambda=100$ | $\lambda=1000$ |
| PSNR | 31.38 | **33.29** | 28.96 |
| SSIM | 0.97 | **0.98** | 0.94 |

Finally, the optimal solution is found by performing various experiments, and the proposed method shows higher performances of PSNR and SSIM compared to other methods. Also, the proposed network generates de-blurred images by eliminating the blur noise properly.

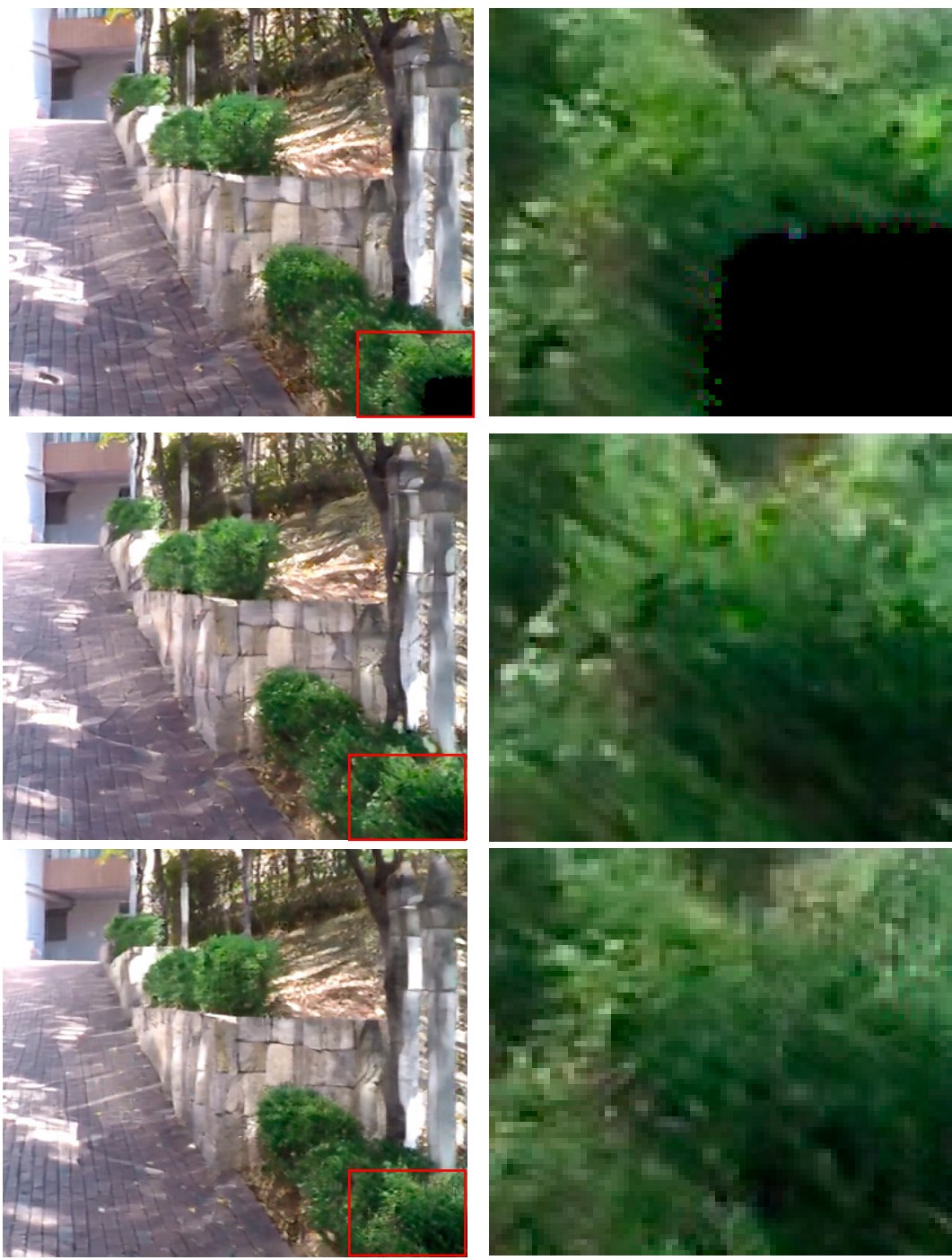

**Figure 22.** Generated images from extracting different layers in VGG16. First row uses low layers, second row combines low layers and high layer, and last row uses high layers.

**Table 8.** PSNR and SSIM of different lambda on Kohler dataset.

| Method | Comparing Between Different Layers | | |
|---|---|---|---|
| | $\lambda=10$ | $\lambda=100$ | $\lambda=1000$ |
| PSNR | 23.31 | **23.29** | 22.45 |
| SSIM | 0.76 | **0.86** | 0.75 |

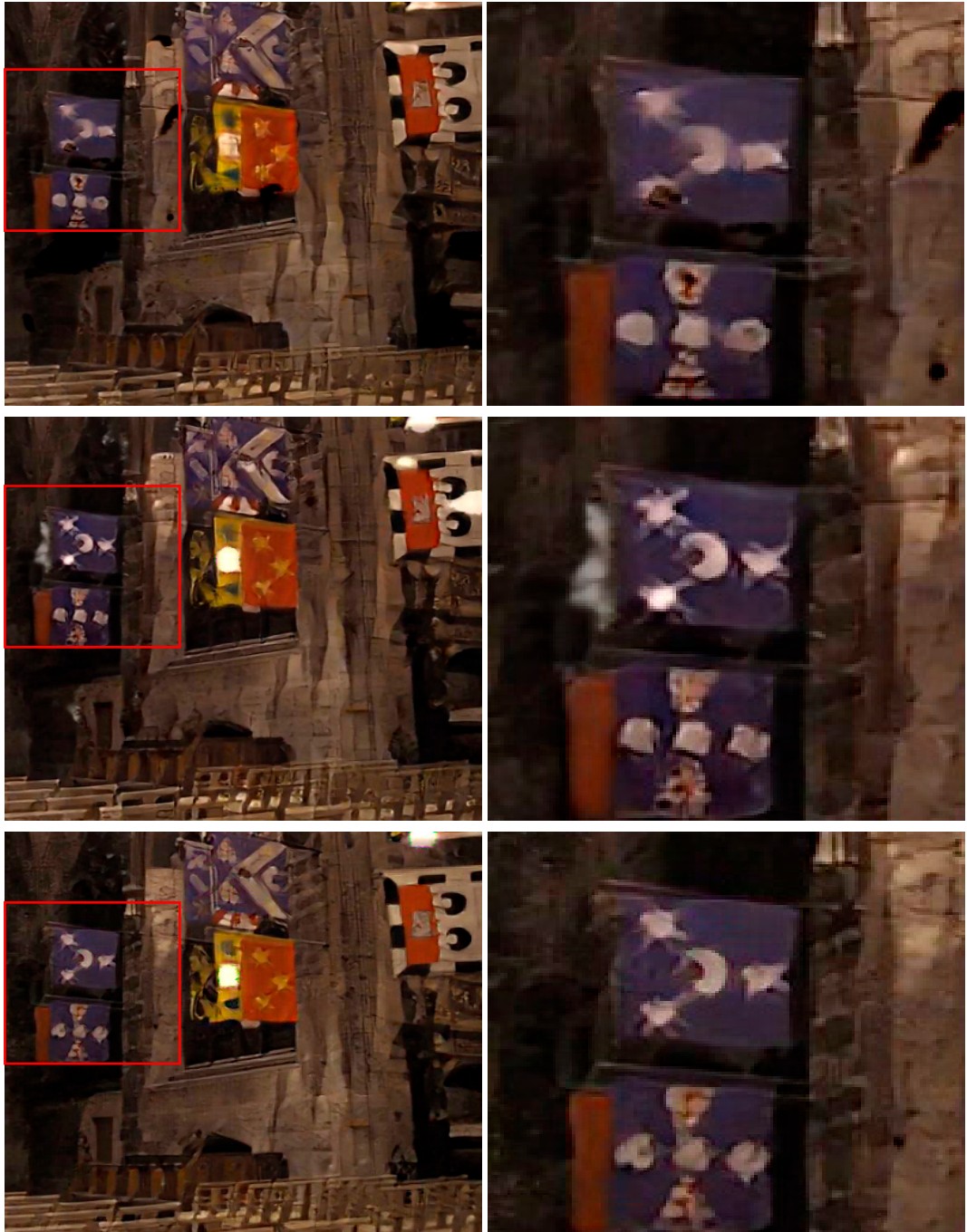

**Figure 23.** Generated images from extracting different layers in VGG16 on Kohler dataset. First row uses low layers, second row combines low layers and high layer, and last row uses high layers.

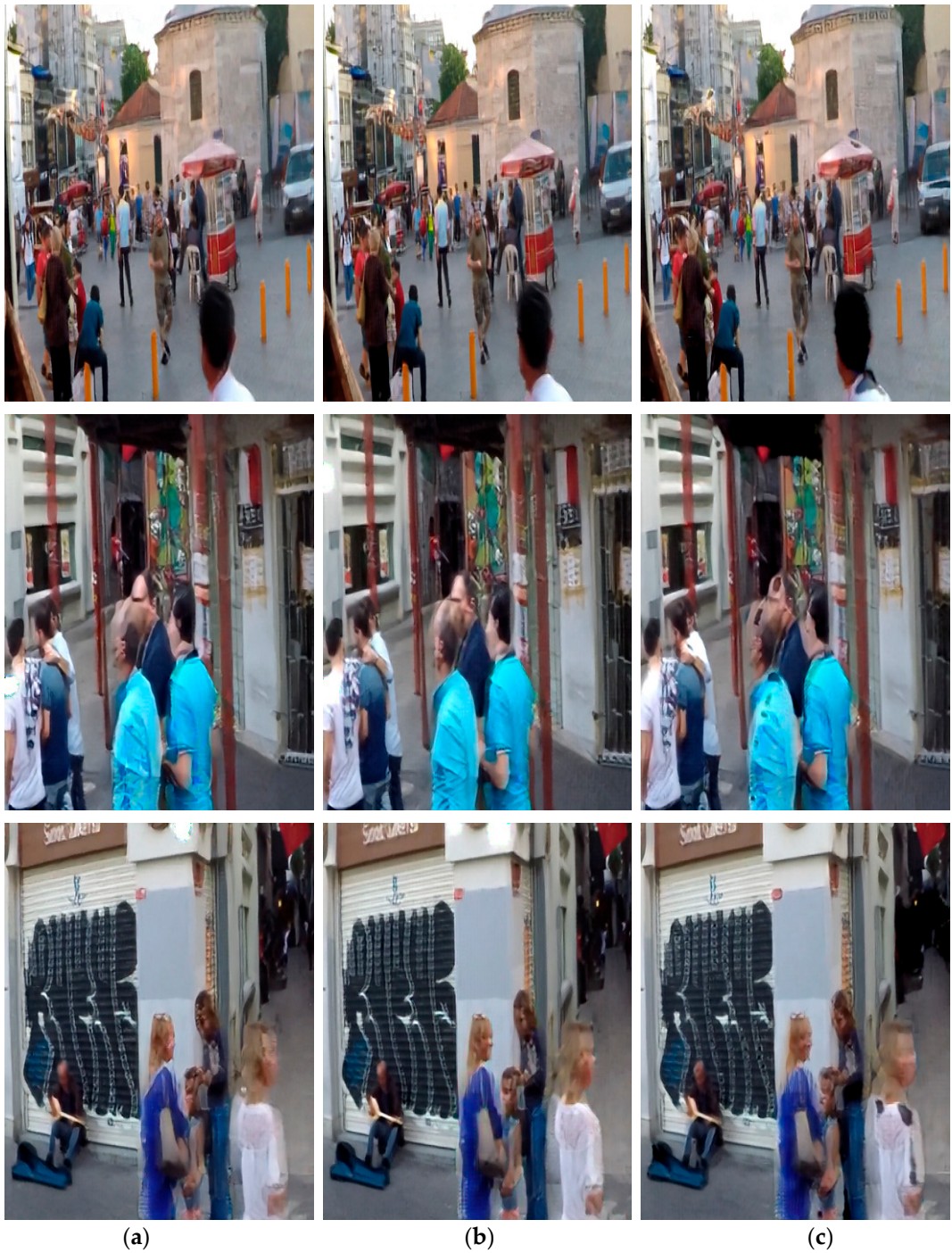

**Figure 24.** Generated object images according to different λ values: (**a**) λ = 10 (**b**) λ = 100 (**c**) λ = 1000.

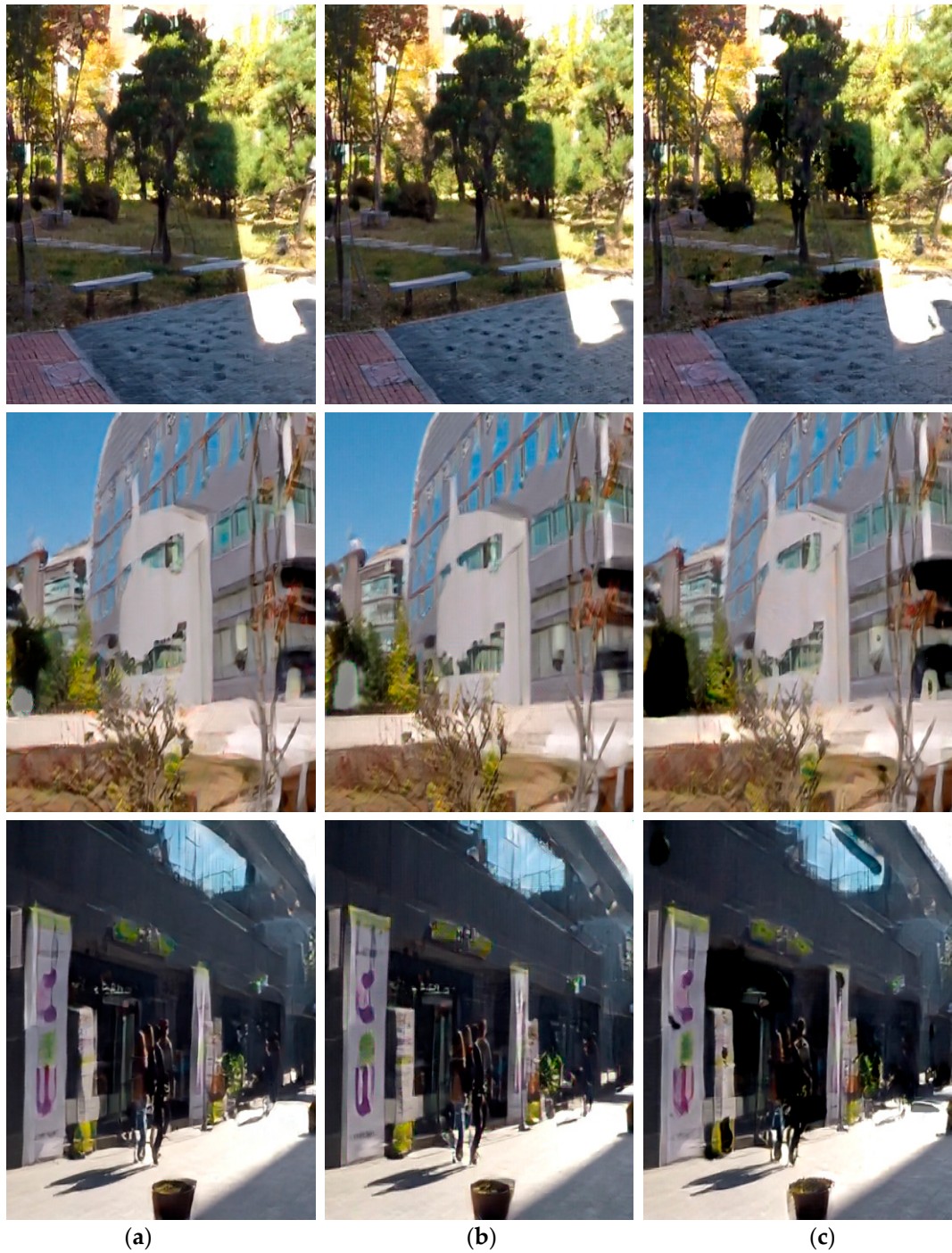

(**a**)       (**b**)       (**c**)

**Figure 25.** Generated background images according to different λ values: (**a**) λ = 10 (**b**) λ = 100 (**c**) λ = 1000.

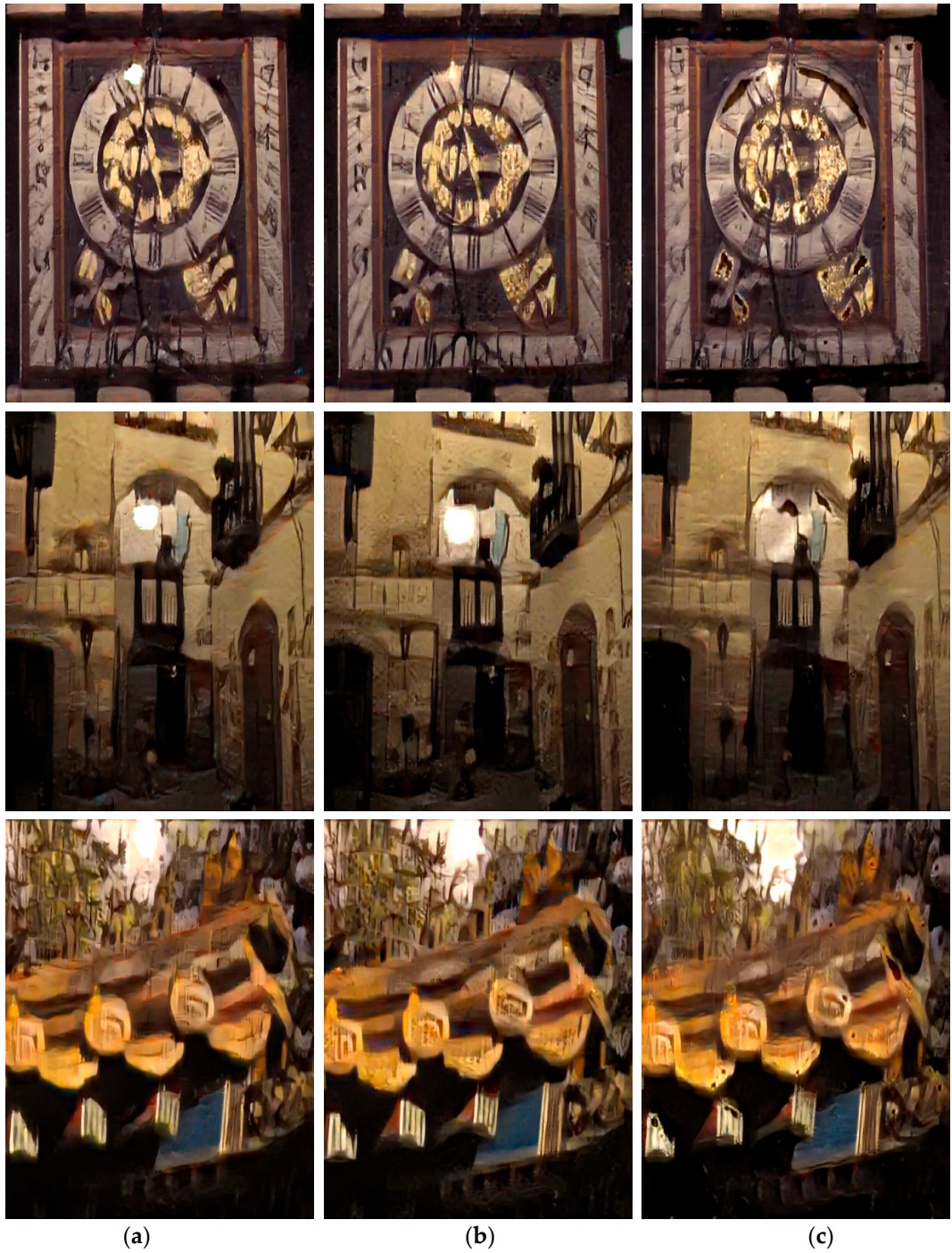

**Figure 26.** Generated images according to different $\lambda$ values on Kohler dataset:. (**a**) $\lambda = 10$ (**b**) $\lambda = 100$ (**c**) $\lambda = 1000$.

## 5. Conclusions

In this paper, a new loss function consisting of the WGAN-GP and style loss function is proposed to reconstruct an edge preserved de-blurred image. The well-trained network, by using the new loss function, is more efficient than other de-blurring methods, and it can improve the similarity between the generated image and the sharp image. The main reason for the improvement in similarity is the application of the style loss function to capture perceptual style. Perceptual style of the sharp image includes sharpness, which means the edge information of the image. Consequently, the network of the proposed method can reconstruct a de-blurred image that contains high frequency components.

In the experiments, the advantage of the proposed method over content-based method was proven through experimental analysis in the case of both the GOPRO Large dataset and Kohler dataset.

The proposed method shows better improvements in PSNR and SSIM performances compared to filter-based methods and the content-based method. Also, a comparative experiment regarding the location of feature maps confirms that the features from high layers are able to preserve edge information. Lastly, it is experimentally suggested that the ratio of WGAN-GP loss to style loss be 1:100 in order to generate the best edge-preserved de-blurred image.

**Author Contributions:** Conceptualization, M.H. and Y.C.; methodology, M.H.; software, M.H.; validation, M.H. and Y.C.; formal analysis, M.H.; investigation, M.H.; resources, M.H.; data curation, M.H.; writing—original draft preparation, M.H.; writing—review and editing, Y.C.; visualization, M.H.; supervision, Y.C.; project administration, Y.C.; funding acquisition, Y.C.

**Funding:** This research received no external funding.

**Acknowledgments:** This work was supported by the Technology Innovation Program (10073229, Development of 4K high-resolution image based LSTM network deep learning process pattern recognition algorithm for real-time parts assembling of industrial robot for manufacturing) funded By the Ministry of Trade, Industry & Energy (MOTIE, Korea).

**Conflicts of Interest:** The authors declare no conflict of interest.

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
