# Peer review of "Wasserstein Generative Adversarial Network Based De-Blurring Using Perceptual Similarity"

_applsci, doi:10.3390/app9112358_

Round 1
Reviewer 1 Report
The manuscript discussed about the use of WGAN based algorithm to de-blur images. The study is interesting and the outcomes could be useful. Some changes are required to be done on the manuscript to make it more readable. Specific comments are listed below.
1) Writing is sometime informal and some grammatical mistakes have been spotted in the manuscript. The authors should thoroughly proof-read the manuscript before submission. In addition, avoid to use short paragraphs (paragraph with just one or two sentences), e.g. line 30-31, 73-74 and others.
2) The writing style is like report writing which is not so appropriate for journal manuscript. The reviewer recommended the authors to change the style of writing in the revised manuscript. In line 81-88, it seems that the term Section is more appropriate than Chapter when describing different sections of the manuscript.
3) Avoid using 'we' in academic writing.
4) Captions in Fig. 12 should be re-arrange as they are partly overlapping.
5) There is a bit of confusion in Fig. 16 thru to Fig. 21. The authors said that you are comparing the image generated by two algorithms. However, it is unclear which photos in these figures referred to which algorithm. The authors should include sub-figure numbering to these figures and the caption in these figures should provide more information in this matter.
6) Avoid starting a sentence with 'And'.
7) In Figures 23 and 24, should the images generated by the proposed mechanism compared with a sharp image to evaluate the ability of the proposed algorithm in de-blurring images?
8) More information should be provided in the captions for Figure 25-27 about what lambda value was used in each figure.
Author Response
Please find the attached responding file.

Reviewer 2 Report
The manuscript presents the application of the Wasserstein Generative Adversarial Network with Gradient Penalty (WGANGP) algorithm in the process of de-blurring images.
The abstract needs some work in terms of structure: Introducing the problem, the motivation to solve it, the aim more clearly, the methodology followed, the main results obtained and their future impact. It may be shorter.
I strongly recommend the authors to replace the personal pronoun “We” throughout the text, also to replace the text “chapter” per “section”, this is not a book.
The Introduction is well written and structured, the methodology well explained and clear.
At line 265, I suggest to change from “Chapter 3…” to “This section…”
Figures 16 to 21 require better explanation, in what each column represents, the same applies to figure 25 to 27.
The technical soundness of the manuscript is good and adequate; the results support the conclusions.
The references are recent and enough for the scope of the presented work.
Author Response

(The authors gave the same response as above.)

Round 2
Reviewer 1 Report
The manuscript has been revised well and I would recommend to accept the manuscript in the present form.